# A Non-functional γ-Aminobutyric Acid Shunt Pathway in Cyanobacterium *Synechocystis* sp. PCC 6803 Enhances δ-Aminolevulinic Acid Accumulation under Modified Nutrient Conditions

**DOI:** 10.3390/ijms24021213

**Published:** 2023-01-07

**Authors:** Simab Kanwal, Wanchai De-Eknamkul

**Affiliations:** Natural Product Biotechnology Research Unit, Department of Pharmacognosy and Pharmaceutical Botany, Faculty of Pharmaceutical Sciences, Chulalongkorn University, Bangkok 10330, Thailand

**Keywords:** γ-aminobutyric acid, δ-aminolevulinic acid, ALA, GABA shunt, C5 pathway, *Synechocystis* sp. PCC 6803

## Abstract

To redirect carbon flux from the γ-aminobutyric acid (GABA) shunt to the δ-aminolevulinic acid (ALA) biosynthetic pathway, we disrupted the GABA shunt route of the model cyanobacterium *Synechocystis* sp. PCC 6803 by inactivating *Gdc*, the gene-encoding glutamate decarboxylase. The generated Δ*Gdc* strain exhibited lower intracellular GABA and higher ALA levels than the wild-type (WT) one. The Δ*Gdc* strain’s ALA levels were ~2.8 times higher than those of the WT one when grown with levulinic acid (LA), a competitive inhibitor of porphobilinogen synthase. Abiotic stress conditions including salinity induced by 10 mM NaCl and cold at 4 °C increased the ALA levels in Δ*Gdc* up to ~2.5 and 5 ng g^−1^ cell DW, respectively. The highest ALA production in the Δ*Gdc* cyanobacteria grown in BG11 medium was triggered by glucose induction, followed by glutamate supplementation with 60 mM of LA, thereby resulting in ~360 ng g^−1^ cell DW of ALA, that is >300-fold higher ALA accumulation than that observed in Δ*Gdc* cyanobacteria grown in normal medium. Increased levels of the *gdhA* (involved in the interconversion of α-ketoglutarate to glutamate) and the *hemA* (a major regulatory target of the ALA biosynthetic pathway) transcripts occurred in Δ*Gdc* cyanobacteria grown under modified growth conditions. Our study provides critical insight into the facilitation of ALA production in cyanobacteria.

## 1. Introduction

Apart from proteinogenic amino acids that form the essential building blocks of proteins, many non-protein amino acids are also produced in animal, plant and microbial cells [1,2,3]. These metabolites take part in essential physiological and biochemical processes in biological systems. Amongst the numerous non-protein amino acids found in prokaryotic and eukaryotic living systems, δ-aminolevulinic acid (ALA) is considered particularly important due to its ubiquitous presence and multifunctional properties in the metabolism of bacteria, algae, plants and animals, as well as due to its use in cosmetic, agricultural and pharmaceutical industry products [4]. ALA is considered a significant intermediate in tetrapyrrole biosynthesis; it plays a critical role in respiration and photosynthesis by controlling chlorophyll metabolism in green organisms [5].

The biosynthesis of ALA inside the cells chiefly occurs via two distinct metabolic routes: the C4 pathway (Shemin pathway) and the C5 pathway (Beale pathway), as illustrated in Figure 1. The C4 pathway appears in mammalian, fungal, protozoan and bacterial cells, in which ALA forms from succinyl-CoA and glycine via a single-step catalytic action of ALA synthase (ALAS) [5]. In the C5 pathway (hereafter referred to as the ‘ALA pathway’), occurring in blue-green algae (cyanobacteria), plants and several bacterial species, ALA forms from glutamate in three irreversible steps and is catalysed consecutively by the action of glutamyl-tRNA synthetase (GluRS), glutamyl-tRNA reductase (GluTR) and glutamate-1-semialdehyde 2,1-aminomutase (GSA-AM) [6]. In both the C4 and the ALA pathways, ALA is further catabolised to porphobilinogen (PBG) via porphobilinogen synthase (Pbs) and is subsequently metabolised to chlorophyll, phycobilins or heme [7]. In photoautotrophic and non-nitrogen-fixing cyanobacterium *Synechocystis* sp. PCC 6803 (hereafter referred to as ‘*Synechocystis*’), the genes encoding the enzymes along the ALA pathway (glutamate → PBG) are *gltX*, *hemA*, *hemL* and *hemB*, with the respective gene IDs: *sll0179*, *slr1808*, *sll0017* and *sll1994* (http://genome.microbedb.jp/cyanobase/GCA_000009725.1 (accessed on 7 April 2022)). On the other hand, glutamate (a precursor metabolite in the ALA pathway) is also a chief substrate for an adjacent metabolic route known as the ‘GABA shunt pathway’ in *Synechocystis* [8]. Previously, ALA production has been achieved via pathway engineering or nutrients optimization in various microbial strains including *Yarrowia lipolytica*, *E. coli* and *Corynebacterium glutamicum* [9,10,11]. However, although *Synechocystis* has been widely used as a model cyanobacterium for research because of its ability to grow heterotrophically on glucose as well as its publicly available genome sequence [12,13], the study on ALA production in *Synechocystis* with reference to GABA shunt pathway is still lacking. The GABA shunt pathway is an alternative yet less energy-efficient metabolic route for completing the tricarboxylic acid (TCA) cycle in a few cyanobacterial species [14]. This metabolic route operates independently of the TCA cycle by converting α-ketoglutarate to glutamate, GABA and succinyl semialdehyde. Comprehensively, initial metabolic routes (from glycolysis to TCA cycle) are mutually shared by the GABA shunt and the ALA pathways in *Synechocystis* and any strategy aiming to inhibit the GABA shunt pathway might result in redirecting the carbon flux via glutamate towards other carbon reserve products, such as ALA. Hence, since both the GABA and the ALA pathways share a common precursor (glutamate), a *Synechocystis* mutant strain lacking a functional *Gdc* (gene ID: *sll1641*) to encode glutamate decarboxylase (GDC), named ‘Δ*Gdc*’ that can accumulate higher intracellular glutamate levels, was employed to analyse the ALA synthesis associated with a disrupted GABA shunt route. This study also summarises the role of key influencing physicochemical factors in regulating ALA metabolism in terms of the relevant metabolites and gene expression profiles observed in the *Synechocystis* sp. PCC 6803 wild-type (WT) and Δ*Gdc* strains. Furthermore, the effects of external nutrients and levulinic acid (LA; a competitive inhibitor of Pbs) supplementation on ALA synthesis in the Δ*Gdc* cyanobacteria were analysed to assess potential ALA production in cyanobacteria, a feature that could be particularly useful for future applications.

## 2. Results and Discussion

### 2.1. Disruption of the GABA Shunt Pathway for the Production of ALA

In few cyanobacteria, including the model species *Synechococcus* PCC7002 and S*ynechocystis*, the TCA cycle closes by an alternative bypass route via the GABA shunt. It is operated by a consecutive conversion of glutamate (chiefly deriving from the TCA cycle metabolite α-ketoglutarate) to GABA, subsequently closing the cycle through a succinyl semialdehyde formation (Figure 1). In *Synechocystis*, the GABA shunt was more active than the widely-distributed TCA cycle route and it was also studied for its possible role in overcoming physiological stresses [3]. The glutamate metabolism at this branch point forms a major connection between the carbon and nitrogen metabolism and provides the major metabolic precursors of the ALA pathway [15]. It has been shown previously that interruption in *Gdc* led to glutamate accumulation in plants and cyanobacteria [16,17]. However, redirecting carbon flux from the GABA shunt to the ALA biosynthetic pathway, in the form of glutamate utilisation, has not been previously studied in cyanobacteria. Hence, considering that glutamate availability to the ALA pathway might be enhanced in the absence of a functional GABA shunt in *Synechocystis*, we generated a mutant strain with a disrupted GABA shunt metabolic route through the inactivation of *Gdc* gene by using the strategy shown in Figure 2a and described in the experimental procedures (Section 3.2). The engineered strain, Δ*Gdc*, was further analysed to determine its growth rate and chlorophyll *a* content. Firstly, comparing the growth patterns of the WT and the Δ*Gdc* strains shows that GDC deactivation does not compromise organism growth, except for the period between day 2 to day 8 (until the establishment of the late-log phase), where the Δ*Gdc* strain exhibits a slightly declined growth (Figure 3a). Later on, during the stationary phase, the growth of the Δ*Gdc* strain was comparable to that of the WT one. Thus, in the absence of a functional GABA shunt, the organism could compete well for nutrient-depletion or pH change in the medium, two substantial factors influencing microbial cell growth during the stationary phase [18]. On the other hand, chlorophyll *a* content in the Δ*Gdc* cyanobacteria, that was about 1.5, 2.5 and 2.4 µg mg^−1^ cells at mid-log, late-log and stationary phases of growth, respectively, was slightly lower than that observed in the WT ones (that was about 1.7, 2.6 and 3 µg mg^−1^ cells at mid-log, late-log and stationary phases of growth, respectively) (Figure 3b), thereby suggesting a clear impact of the disrupted GABA shunt on the C5 pathway of *Synechocystis*. As depicted in Figure 1, the ALA catabolism bifurcates into two metabolic routes, i.e., chlorophyll *a* or heme formation, so the decrease in chlorophyll *a* might be possibly due to increased flux to heme formation that is needed to investigate further as the ALA flux to chlorophyll *a* or heme synthesis is unknown in Δ*Gdc* yet. These results indicate that despite the decreased photosynthetic pigment levels, the Δ*Gdc* strain had efficient carbon assimilation to maintain its growth under normal photoautotrophic conditions.

Under normal growth conditions (untreated), the maximal ALA levels in the *Synechocystis* mutant strain were about 1 ng g^−1^ DW of cells, that is approximately 7 times higher than those observed in the WT one at the late-log growth phase (after 6 days of culturing; Figure 4a). ALA accumulation was enhanced in both the WT and the Δ*Gdc* strains in response to modified growth conditions. In an attempt to impose the smallest effect on the growth of cyanobacteria, a two-stage culture approach was established. The cells were first pre-grown under normal photoautotrophic conditions until their growth rate was comparable (at the late-log phase). They were subsequently subjected to the modified physicochemical conditions to generate high ALA levels. Numerous attempts have been made to enhance the feasibility and efficiency of ALA production in bacteria and fungi by modifying the nutrient supply or the growth conditions [19,20]. Among the nutrient factors, the presence of carbon or inorganic nitrogen sources, mineral salts, amino acids and ions, as well as the concentrations of precursors or inhibitors, were found to affect ALA production. There are several reports regarding the stress-relieving roles of ALA in plants in response to certain environmental factors (including abiotic signals such as salinity and drought) [21,22]. As a result, we examined whether various physicochemical factors could lead to an elevated ALA accumulation in the Δ*Gdc* mutant. We observed that the cellular ALA contents were higher in the cold-treated WT strain and increased significantly in the Δ*Gdc* strain compared to those in untreated cells. Moreover, salinity induced osmotic stress led to a 2-fold increase in the ALA levels in the examined *Synechocystis* mutant. As already mentioned, the exogenous ALA application can confer stress tolerance in higher plants; however, limited information is available regarding any such role of ALA in cyanobacteria or in the regulation of ALA metabolism and the in vivo synthesis of ALA, except for some research work that have been done decades ago by Beale [23], Troxler and Brown [24], Kipe-Nolt et al. [25] and by Rieble and Beale who have conducted studies on the characterisation of the GluTR in *Synechocystis* [26]. The effects of osmotic stress on the upregulation of the GDC pathway at the enzymatic activity and the transcription level in *Synechocystis* have been previously reported [27]. In this study, the increased ALA levels might be a response to the metabolic perturbations occurring in the absence of a GDC activity and the stress caused by the applied cold and salinity conditions. Transcriptional analyses were undertaken to obtain more information regarding the possible effect of those abiotic factors on gene regulation in the ALA pathway.

We showed that external nutrients, such as glucose and glutamate, can increase the ALA levels in the Δ*Gdc* strain up to ~3 and 2.1 times, respectively, in 24 h, compared to the cells grown in a normal medium. The increased duration of the exposure to the aforementioned nutrients (for up to 48 h) led to a further increase in the observed ALA accumulation. *Synechocystis* cells can uptake exogenous glucose as an efficient carbon source [28] and a major proportion of the assimilated carbon is known to be consumed by the TCA cycle metabolism (that is also a source of precursors for the GABA shunt and the ALA pathway) [29]. Glucose is the best carbon source for enhancing the growth and GABA levels of *Synechocystis* cells by upregulating the GDC transcript levels and catalytic activity [27]. On the other hand, inexpensive glucose has also been used as an efficient sole carbon source of the C5 pathway of bacteria for ALA production [20]. There are also reports of ALA overproduction via C5 pathway in various genetically engineered microorganisms grown under glutamate supplementation and achievement of higher ALA titers as a result of flux redistribution of the TCA cycle toward glutamate [30,31]. Further, we have observed that a *Synechocystis* mutant strain over expressing *hemA* requires glucose or glutamate supplementation essentially for its growth (data not published), hence complementing the results in present study that both nutrients serve as precursor substrates for ALA biosynthesis and the increase in ALA titers observed in the Δ*Gdc* cyanobacteria were likely due to the availability of more substrate, owing to a lack of glucose and glutamate utilisation by the disrupted GABA shunt pathway. Succinate (a precursor for ALA synthesis in bacteria) did not seem to affect ALA production in *Synechocystis*. In contrast, there was a minor increase in ALA levels in glycine-treated *Synechocystis* cells compared to untreated cells. Glycine serves as the chief substrate for ALA synthesis via the ALAS catalysis in the C4 pathway (as described in introduction section and shown in Figure 1) and is largely restricted to a few bacteria, fungi and mammals and has not been demonstrated in plants and cyanobacteria. *Synechocystis* possesses the glycine uptake and degradation system [32]. Yet, there is no evidence of an ALA biosynthesis from glycine in cyanobacterial cells and there has been no direct synthesis link established between glycine and the ALA precursor glutamate. Hence, we assumed that the altered ALA levels observed in this scenario might result from the regulatory mechanisms of the C5 pathway aroused in response to the toxic cellular effects of glycine [32,33].

We subsequently analysed the effect of GDC inactivation on cellular metabolites that have a direct metabolic link to both the GABA shunt and the ALA pathway. We observed a significant increase and decrease in the cellular glutamate and GABA contents, respectively, in the Δ*Gdc* strain (as compared to those of the WT one) (Figure 4b). Previous studies showed that a certain ATP-binding cassette-type transporter facilitates the glutamate uptake of *Synechocystis* cells. This glutamate is further utilised in various cellular metabolic processes, including the glutamine synthetase and glutamate synthase (GS/GO-GAT) cycle and the GABA shunt pathway, which are relevant to the interchangeable α-ketoglutarate and glutamate contents, respectively [34,35]. Fresh water cyanobacteria grow ideally under slightly alkaline pH conditions and tend to maintain their intracellular pH within the range of approximately 6.9 to 7.8 [36,37]. In this scenario GABA shunt pathway is one among other pathways that plays important role in buffering the cytosolic pH of cyanobacterial cells by maintaining the levels of acidic compound such as glutamate that is proved to be toxic to cells at higher concentration (>30 mM) and impedes cyanobacterial growth [8,38]. Our results indicate that an external glutamate supply had helped increase intracellular glutamate levels in the Δ*Gdc* cyanobacteria. This glutamate is also further converted to α-ketoglutarate by glutamate dehydrogenase (GdhA) to regenerate the nicotinamide adenine dinucleotide phosphate (NADPH) required for ALA synthesis (Figure 1). These results were further supported by transcriptional analysis of relevant genes in these pathways and are discussed in the next section.

### 2.2. Transcriptional Regulation of the ALA Biosynthesis Pathway in the ΔGdc Mutant Strain

In an attempt to gain a better understanding of the underlying biology behind the effect of the GDC inactivation on the ALA biosynthesis pathway, we quantified the mRNA expression of *gltX*, *hemA*, *hemL*, *hemB*, *Gdc* and *GdhA* in the WT and the Δ*Gdc* strains, under normal and modified growth conditions. The results demonstrated that the transcript levels of genes of the ALA pathway were upregulated in the Δ*Gdc* strain compared to the WT one. In contrast, certain modified growth factors had substantially induced the expression of each gene of the ALA biosynthesis pathway in *Synechocystis* (Figure 5a). While comparing the effects of two different abiotic stress conditions on the mRNA levels of these genes, we found that the cold condition had increased the expression of *gltX*, *hemA*, *hemL* and *hemB* in the WT strain compared to the osmotic stress condition. The expression of those genes was further enhanced in the Δ*Gdc* cyanobacteria, consistent with the observed increased ALA production (Figure 4a). We observed that both glucose and glutamate increased the expression of *gltX*, *hemA*, *hemL* and *hemB* several times in the Δ*Gdc* strain compared to the control condition. According to previous studies, overall, the ALA metabolic pathway is regulated at various steps in response to changes in gene expression and intermediate products of the downstream metabolic routes [7]. Based on the transcriptional and enzymatic analysis performed in plants and bacteria, *hemA* and *hemB* (encoding GluTR and Pbs, respectively) are the major regulatory targets of the ALA biosynthesis pathway [7,39]. Since ALA is not the end-product in this metabolic route, GluTR (that catalyses the rate-limiting step in ALA synthesis) is feedback-inhibited by the end-product (heme). On the other hand, Pbs (that catalyses the major ALA catabolic reaction) is also feedback-inhibited by the downstream intermediate protoporphyrinogen IX in the heme biosynthesis pathway, while the overexpression of *hemB* has adversely affected ALA accumulation. LA, a structural analogue of ALA, can act as a competitive inhibitor of Pbs and has been used in many biotechnological studies to trigger ALA accumulation in microbes [23,40]. Hence, both *hemA* and *hemB* act as key regulatory nodes of the ALA metabolism. Furthermore, we observed that the glycine supplementation had intriguingly upregulated the transcript amounts of all four genes involved in the ALA metabolic pathway in both the WT and the mutant strains, in contrast to those of the control growth condition. As in higher plants, the transcriptional and post-transcriptional regulatory mechanisms as well as posttranslational enzymatic regulations of important metabolic processes in *Synechocystis* have been indicated in several studies [41,42]. Although ALA accumulation did not increase significantly after a 24 h glycine treatment (Figure 4a) in this study, the obvious transcriptional alterations of the pathway genes indicate a post-transcriptional regulation or feedback-inhibition of the ALA pathway by the downstream intermediate products that will be worth investigating in future.

Interestingly, the Δ*Gdc* cells contained higher transcript levels of *gdhA* (the gene that encodes GdhA) in response to cold, glucose, glutamate and glycine supplementation when compared to those of the WT ones. In cyanobacteria, the carbon/nitrogen homeostasis is attained mostly by regulating the metabolite levels of α-ketoglutarate through various enzymatic pathways, including the GdhA (which catalyses the reversible conversion of α-ketoglutarate and glutamate) [43]. Moreover, the GdhA catalysed reaction acts as a connecting link between the TCA cycle and the ALA pathway. The upregulation of the *gdhA* transcription in the Δ*Gdc* cells grown under modified nutrients and abiotic stress is, hence, another finding that acts as evidence of the glutamate flow to neighbouring pathways when the GABA shunt pathway is not functioning in cyanobacteria. The inactivation of *Gdc* had no apparent effect on the pathway genes in the *Synechocystis* that was exposed to succinate supplementation.

The above analysis suggested that the upregulation of *hemB* might have increased Pbs activity, consequently limiting ALA accumulation. We subsequently investigated the effect of various LA concentrations on the expression levels of pathway genes in *Synechocystis* WT and Δ*Gdc* strains. As expected, a significant increase in the transcription of ALA biosynthesis genes, especially of *hemA* and *hemL*, was observed in *Synechocystis*, with the highest mRNA levels observed in Δ*Gdc* cells in the presence of 60 mM LA (as shown in Figure 5b). Moreover, the transcript levels of *gdhA* were also markedly increased in *Synechocystis* in the presence of 60 mM LA, whereas the *Gdc* transcripts were not considerably upregulated in WT cells exposed to 60 mM LA, thereby suggesting that the LA inhibition of Pbs results in an upregulation of the GdhA and the ALA synthesis route to benefit the ALA accumulation. However, the mRNA levels of *hemB* (whose product is inhibited by LA) were increased in LA-treated cells, indicating the inadequate feedback-inhibition occurring from the downstream metabolic products. This inadequacy is due to an interruption in the ALA catabolism that results in an upregulation of *hemB*, but not of the activity of Pbs, as LA blocks the latter. This biochemical change might also be responsible for the increased *hemA* and *hemL* transcription, whose products are directly involved in the catalysis of the ALA biosynthesis. The effect of LA on the cellular ALA levels of *Synechocystis* was assessed to validate these results further.

**Figure 5 ijms-24-01213-f005:**
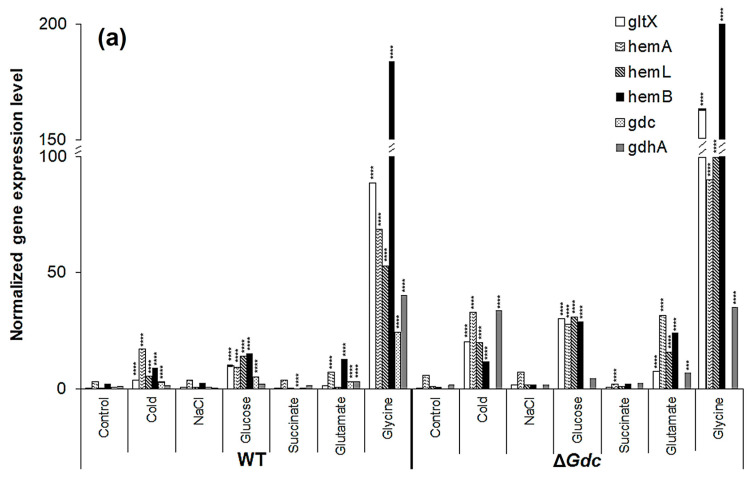
Transcriptional regulation of pathway genes in the *Synechocystis* wild-type (WT) and the *Gdc* knockout mutant (Δ*Gdc*) strains. Analysis of the gene expression profile of various genes related to δ-aminolevulinic acid (ALA) and glutamate metabolism in *Synechocystis* sp. PCC6803 (WT) and the Δ*Gdc* strain grown under (**a**) various treatments and (**b**) different concentrations of levulinic acid (LA). Results were normalised to 16S rRNA levels and bars represent the mean ± standard errors of three independent real-time RT-PCR reactions. If the error bar cannot be seen, the deviation is very small. *** *p* < 0.01, **** *p* < 0.001 versus untreated group. The respective gene IDs and the descriptions of the targeted genes are provided in Table 1 and Figure 1.

### 2.3. Cyanobacterial Growth and ALA Production in the Presence of LA

We investigated the effect of LA on the ALA synthesis and the growth of cyanobacteria cultured under normal photoautotrophic conditions. From the results shown in Figure 6a,b, it is obvious that LA at higher concentrations (40–60 mM) had significantly impeded the cell growth (biomass levels) of both the WT and the Δ*Gdc* mutant strains, as there was a continuous decline in biomass levels within 48 h of the LA supplementation. The maximum amount of ALA was produced in the presence of 60 mM of LA in the *Synechocystis* cultures. In cultures supplemented with a lower LA concentration (20 mM), the growth of the Δ*Gdc* cells was affected more than that of the WT ones, yet no increase in the ALA content was observed. Our results are supported by previous studies reporting the growth inhibitory effect of LA supplementation on microalgae, which is a possible consequence of the inhibition of chlorophyll synthesis [23,25]. Previously, it was noticed that cell growth of cyanobacterial cultures was completely inhibited in response to LA concentrations above saturation point and also that just 10 mM LA caused nearly 50% inhibition in chlorophyll synthesis of an algal culture [23,25]. In this regard, further studies are needed to be undertaken in future to observe the saturation-type kinetics of ALA production at different LA concentrations in *Synechocystis*. In order to further validate the notion that ALA accumulation in the glucose- or the glutamate-supplemented Δ*Gdc* cells may be limited owing to a higher Pbs activity that further breakdowns ALA into downstream metabolites, the levels of ALA were quantified in LA-treated Δ*Gdc* cyanobacteria grown in a medium that was supplemented with 10 mM of glucose or glutamate. The LA treatment benefited the ALA accumulation in Δ*Gdc* cells, thereby resulting in a nearly 6- and 7-fold increased ALA content in the presence of glutamate and glucose (that were correspondingly ~290 and 360 ng g^−1^ cell DW), respectively, as compared to the ALA content in Δ*Gdc* cells treated with LA alone (Figure 6c). Bioproduction of ALA has been achieved previously from harmless and non-pathogenic microorganisms including *Yarrowia lipolytica*, *E. coli* and *Corynebacterium glutamicum*, with the maximum ALA yield reaching 2.217, 11.5 and 1.79 g L^−1^, respectively [9,10,11]. Although the ALA production in cyanobacterial cells from current study is far less than the other studies reported for photosynthetic microorganisms (Table 2), our results suggest that optimising ALA production in *Synechocystis* could be achieved by balancing the objective metabolic routes by targeting the metabolic engineering and precursors/inhibitors.

We also considered that the ALA biosynthesis in *Synechocystis* is tightly regulated at both the transcriptional and the post-transcriptional levels in response to the fluctuations in neighbouring/competing pathways, external growth factors and the downstream intermediates’ flux. By looking into the important regulatory role of *hemB* in ALA accumulation in *Synechocystis*, a detailed characterisation of its product would be desirable, along with further research on the inhibition kinetics caused by downstream products, LA and other protein inhibitors; such an insight might prove beneficial for the utilisation of cyanobacteria as cell factories for ALA biosynthesis, particularly from the perspective of their utility in future applications.

## 3. Materials and Methods

### 3.1. Strains and Culture Conditions

The WT strain of *Synechocystis* sp. PCC 6803 was obtained from the Pasteur culture collection (Paris, France) and the Δ*Gdc* mutant strain was grown in BG11 medium supplemented with Na_2_CO_3_ (BDH, Poole, Dorset England) as a carbon source and 20 mM HEPES-NaOH (pH 7.5) (Sigma-Aldrich, Saint Louis, USA) [47] in 500 mL cotton-plugged Erlenmeyer flasks with a working volume of 200 mL [17,27]. Kanamycin (25 μg mL^−1^) was added into the BG11 medium for the growth of the Δ*Gdc* strain. Cells were cultured at 28 °C ± 2 °C under continuous illumination at 50 μmol photons m^−2^ s^−1^. Illumination was provided by means of cool white LED lamps (LUMAX Ecosaveplus LED, Shanghai, China) from two sides and the growth rate was determined by measuring the optical density (OD) at 730 nm (OD_730_). Cultures were maintained on solid agar plates containing BG11 media that were supplemented with 10 mM TES (pH 7.5) (Sigma-Aldrich, Saint Louis, USA), 3 g L^−1^ Na-thiosulfate (Ajax Finechem, Sydney, Australia) and 1.5% agar (Difco, Sparks, USA), with or without antibiotic for the growth of the mutant or the WT strain, respectively. The *E. coli* strain DH5α (stored in the lab) was a host for constructing and storing the recombinant plasmids used in this study. Table 3 summarises all constructs and strains used in this study.

### 3.2. Construction of the *Δ*Gdc Strain

In order to construct the plasmid to inactivate the *Gdc* in the *Synechocystis* genome, the 1.4-kb gene of the *Synechocystis Gdc* was amplified by polymerase chain reaction (PCR) with gene-specific primers (Macrogen, Seoul, Korea), as shown in Table 1. The PCR product was ligated into a pGEM^®^-T Easy Vector (Promega), creating a pGdc plasmid containing the *Gdc* gene. In order to knockout *Gdc*, a 1.2-kb kanamycin-resistant gene from the pUC4K vector was cloned into the *Kpn*I site of the pGdc, producing a pGdc*_kan_* plasmid. The design and the construction of the resulting plasmid pGdc*_kan_* is shown in Figure 2a,b. The obtained plasmid was used to transform *Synechocystis* WT cells by natural transformation [48] and generate the mutant strain with the disrupted *Gdc* (Δ*Gdc*). The transformant cells were selected on a BG11-containing plate supplemented with kanamycin at a concentration of 25 μg mL^−1^, within three weeks. To ensure the insertion and complete segregation of genes, transformants were subcultured for up to five generations. The complete segregation of the obtained *Gdc* knockout was analysed by a colony PCR (Figure 2c and Appendix A).

### 3.3. Experimental Treatments and Analysis of Metabolites

The chlorophyll *a* content in the *Synechocystis* WT and Δ*Gdc* strains was determined by an *N,N* dimethylformamide (DMF) (EMSURE^®^ ACS) extraction, according to a protocol adapted from Jantaro et al. [49]. Briefly, liquid cultures of the WT and Δ*Gdc* strains were initiated at an OD_730_ of ~0.1 and were grown until the mid-log (OD_730_ of 0.3–0.4), late-log (OD_730_ of 0.8–1.0) or stationary (OD_730_ of 1.5–2.0) growth phases. Cell suspensions (corresponding to 10 mg of biomass) from various growth phases were centrifuged at 2500× *g* for 10 min and cell pellets were extracted by using DMF and by vigorous mixing at room temperature, followed by a measurement of their chlorophyll *a* content.

Cellular ALA, glutamate and GABA were extracted from the pelleted liquid cultures of the WT and the Δ*Gdc Synechocystis* strains grown until the late-log phase in 200 mL of BG11 medium, followed by various physicochemical treatments for 24 and 48 h. The physicochemical treatments included supplementing the BG11 medium with 10 mM of NaCl, glutamate, glucose, glycine or succinate (Sigma-Aldrich, Saint Louis, MO, USA) or exposing the cultures to a cold temperature (4 °C). LA (Sigma, Aldrich) was prepared as a stock solution of 1 M and neutralized to pH 7.6 with NaOH (Ajax Finechem, Sydney, Australia). The filter sterilized (0.45-µm Millipore filter) LA solution was added at various concentrations (20, 40 and 60 mM), for 48 h, in late-log phase grown cultures. The extraction and quantification of cellular ALA, glutamate and GABA contents were performed as previously described [44]. In brief, ALA, glutamate and GABA were extracted from dry cells, derivatised with o-phthalaldehyde or 9-fluorenylmethyl chloroformate (Agilent, California, USA) and analysed by an HPLC system (Shimadzu Scientific In-struments Inc., Columbia, IN, USA) at 262 and 338 nm. The quantification of ALA and glutamate was done by comparing the peak areas with standards (purity > 99%; Sigma-Aldrich, Saint Louis, USA). All experiments were performed in triplicate.

### 3.4. Total RNA Extraction and Gene Expression Analysis

*Synechocystis* WT and Δ*Gdc* strains were grown under the experimental conditions described earlier to undertake the ALA extraction step. Cells (at an OD_730_ of ~0.8–1.0) treated under various physicochemical conditions for 24 h were harvested by centrifugation at 2500× *g*, for 10 min, at 25 °C. The cell pellet was used for RNA extraction using GenUP™ Total RNA Mini Kit (biotechrabbit, Berlin, Germany) and reverse transcription-quantitative PCR (RT-qPCR) by using the methods described before [44]. The cDNAs of the WT and the Δ*Gdc* strains were synthesized using RevertUP™ reverse Transcriptase (biotechrabbit), according to the manufacturer’s protocol and used to determine the transcript levels of the genes targeted by corresponding primers (Macrogen, Seoul, Korea); both the primer sequences and the target sequence IDs are shown in Table 1. Experiments were performed by using three independent RT-PCR/qPCR assays.

### 3.5. Statistics

Unless otherwise stated, the data presented represent independent experimental triplicates of the mean and are presented as mean ± standard deviation (SD). The differences between means of the individual groups were analyzed using one-way analysis of variance (ANOVA) via GraphPad Prism 9.3.1 software (San Diego, CA, USA). * *p* < 0.05, ** *p* < 0.01 *** *p* < 0.005, **** *p* < 0.001 versus control groups.

## 4. Conclusions

To the best of our knowledge, increased ALA production in cyanobacterium *Synechocystis* sp. PCC 6803 with disrupted GABA shunt route via *Gdc* inactivation (Δ*Gdc*) has not been previously reported in cyanobacteria. The Δ*Gdc* mutant had increased cellular levels of precursor metabolites for ALA biosynthesis which may then maximize the pathway flux towards ALA synthesis. The carbon flux redirected from GABA shunt pathway towards ALA biosynthesis was further improved under specific nutrients conditions in engineered strain. However, further research regarding engineering of the key regulatory metabolic steps, characterization of enzymes involved and analysis on extracellular ALA secretion shall provide insights into the detailed mechanism of ALA production in *Synechocystis*, that might prove beneficial in utilizing cyanobacteria as cell factories for ALA biosynthesis from the perspectives of potential applications in agriculture as an herbicide, insecticide and a growth-promoting factor for plants and medical field for cancer treatment and other clinical uses.

## Figures and Tables

**Figure 1 ijms-24-01213-f001:**
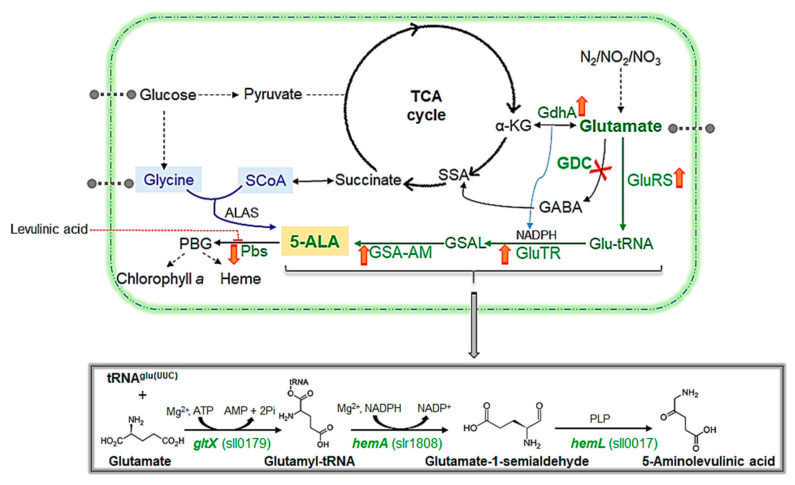
Schematic summary of the ALA biosynthetic pathways in prokaryotes. The C4 pathway (Shemin pathway) is marked in blue, while the C5 pathway (Beale pathway in *Synechocystis*) is marked in green. The native enzymes relevant to the ALA metabolism in *Synechocystis* are marked in green. The pathway engineered in this study is indicated with a red ‘X’ that symbolises the effect of the gene knockout. Cascade reactions are indicated with dashed arrows. The ALA pathway in *Synechocystis* comprises three enzymes: GluRS, GluTR and GSA-AM that are encoded by *gltX*, *hemA* and *hemL*, respectively. These enzymes facilitate the irreversible catalysis of glutamate to ALA, as shown in the overall reaction in the bottom box, along with the relevant genes and their IDs (in green). ALA is converted into PBG by the action of Pbs or of ALA dehydratase (*hemB*, *sll1994*), which is the main enzyme inhibited by levulinic acid (as indicated with a red dotted arrow). The red ‘up’ and ‘down’ arrows beside the gene indicate the up- or downregulation of the genes in the Δ*Gdc* strain, resulting in ALA accumulation. Abbreviations used: 2Pi, diphosphate; α-KG, α-ketoglutarate; ALA, δ-aminolevulinic acid; ALAS, ALA synthase; ATP/AMP, adenosine tri(mono)phosphate, GDC, glutamate decarboxylase; GdhA, glutamate dehydrogenase; GluRS, glutamyl-tRNA synthetase; GluTR: glutamyl-tRNA reductase; Glut-tRNA, glutamyl-tRNA; GSA-AM, glutamate-1-semialdehyde 2,1-aminomutase; GSAL, glutamate-1-semialdehyde; NADPH, nicotinamide adenine dinucleotide phosphate; PBG, porphobilinogen, Pbs, porphobilinogen synthase, PLP, pyridoxal-5ʹ-phosphate; SCoA, succinyl coenzyme-A; SSA, succinic semialdehyde. Reproduced from [8,15,16].

**Figure 2 ijms-24-01213-f002:**
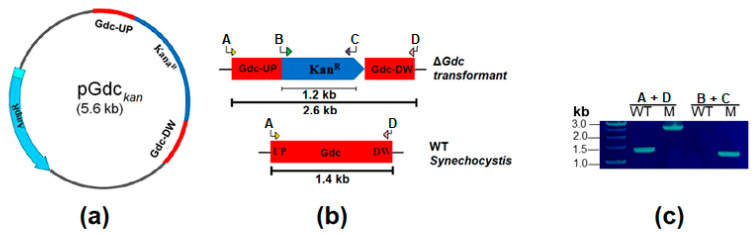
Inactivation of *Gdc* in *Synechocystis*. (**a**) Design and plasmid construction for *Gdc* interrupted by a kanamycin antibiotic-resistant cassette and cloning into a pGEM-T easy vector. AmpR stands for the ampicillin-resistant cassette in pGEM-T easy vector backbone. (**b**) Simplified scheme of the wild-type (WT) *Synechocystis* and of the *Synechocystis* mutant (M) strain with the inactivated *Gdc*. Primers used for the performance of PCR are indicated (arrows labelled ‘A’–‘D’), with the expected PCR product sizes shown as bars beneath both maps. (**c**) Confirmation of the complete segregation of the mutant strain through PCR amplification. Lane WT stands for the wild-type, while lane M stands for the Δ*Gdc* strain, with primer positions as indicated in (**b**).

**Figure 3 ijms-24-01213-f003:**
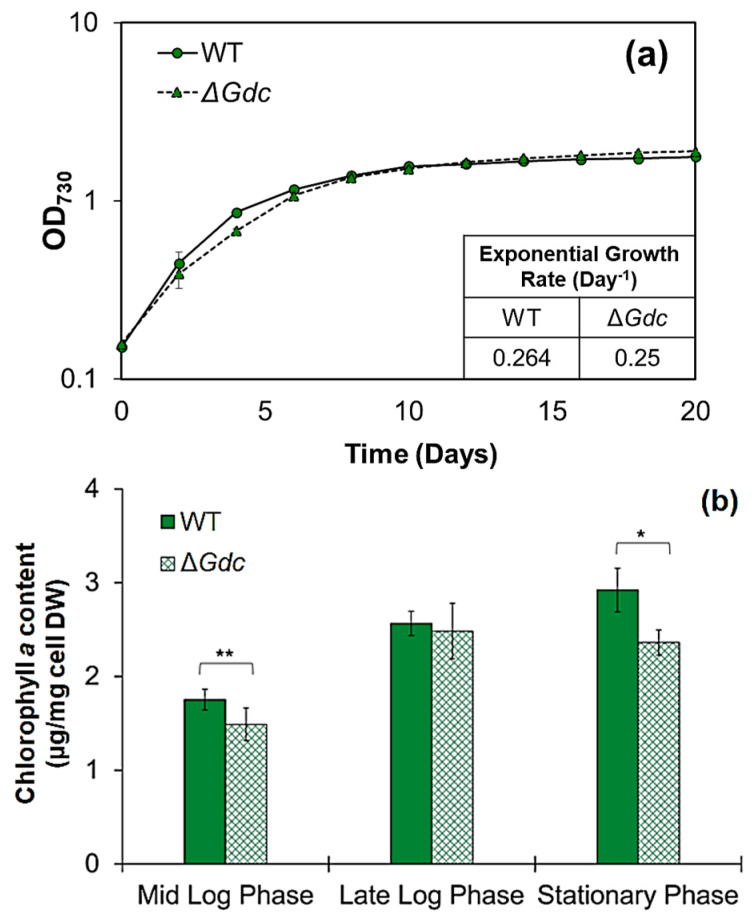
Growth and pigment analysis. (**a**) Comparison of growth by measuring the optical density of cultures at 730 nm. (**b**) Chlorophyll *a* content at various growth phases of the *Synechocystis* wild-type (WT) and mutant (M) strain, grown in BG11 medium and in BG11 medium supplemented with kanamycin (25 µg mL^−1^), respectively. Results represent the mean ± standard deviation (SD) of experiments conducted in triplicate (*n* = 3). The error bars denoting the SD values in (**a**) cannot be seen since the SD is smaller than the size of the symbol. In (**b**), the differences between means of the individual groups were analysed using one-way analysis of variance (ANOVA) via GraphPad Prism 9.3.1 software (San Diego, CA, USA). * *p* < 0.05, ** *p* < 0.01, versus WT cells.

**Figure 4 ijms-24-01213-f004:**
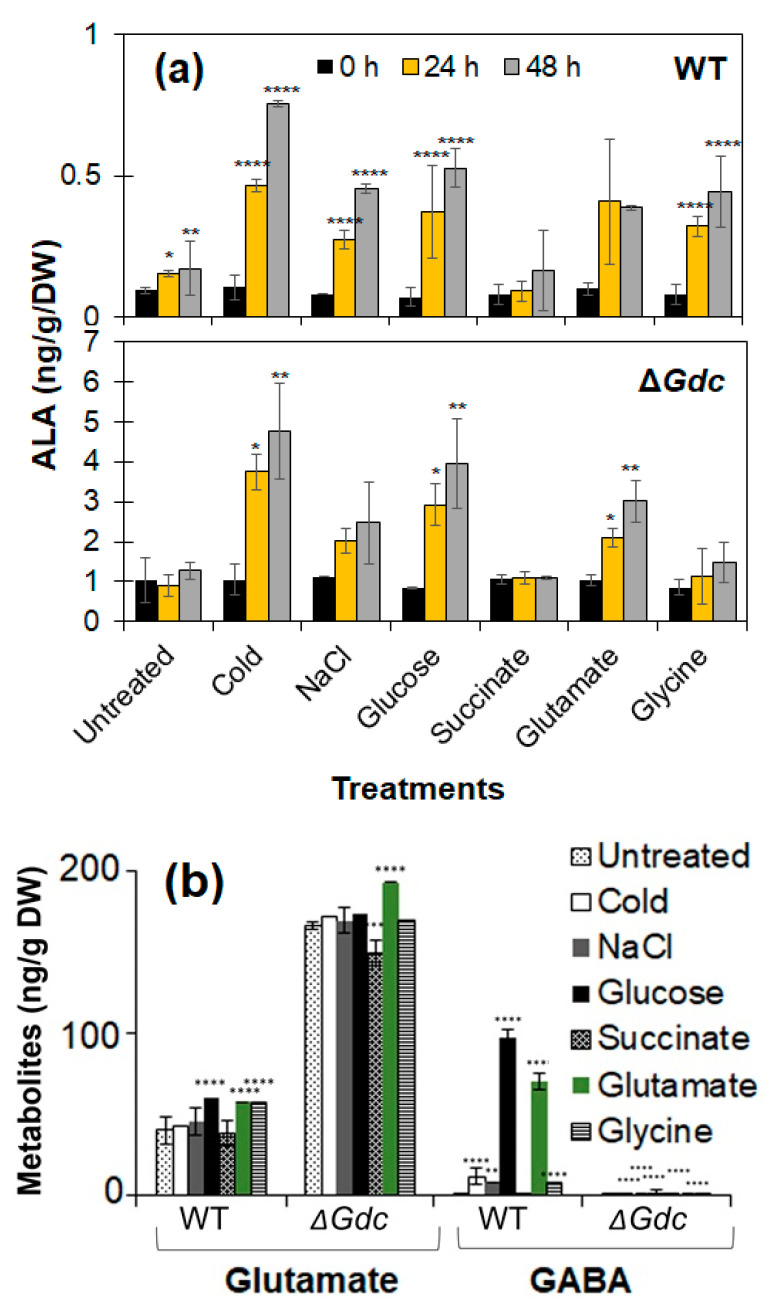
Analysis of the metabolite alteration in *Synechocystis* strains under various treatments. (**a**) δ-Aminolevulinic acid (ALA) and (**b**) relevant metabolite (glutamate and GABA) levels in wild-type (WT) and Δ*Gdc* strains of *Synechocystis*, after using the BG11 medium as a control (untreated). Results represent the mean ± standard deviation (SD) of experiments conducted in triplicate (*n* = 3). The differences between means of the individual groups were analysed using one-way analysis of variance (ANOVA) via GraphPad Prism 9.3.1 software (San Diego, CA, USA). In (**a**), * *p* < 0.05, ** *p* < 0.01, **** *p* < 0.001 versus 0 h, and in (**b**), *** *p* < 0.005, **** *p* < 0.001 versus untreated group.

**Figure 6 ijms-24-01213-f006:**
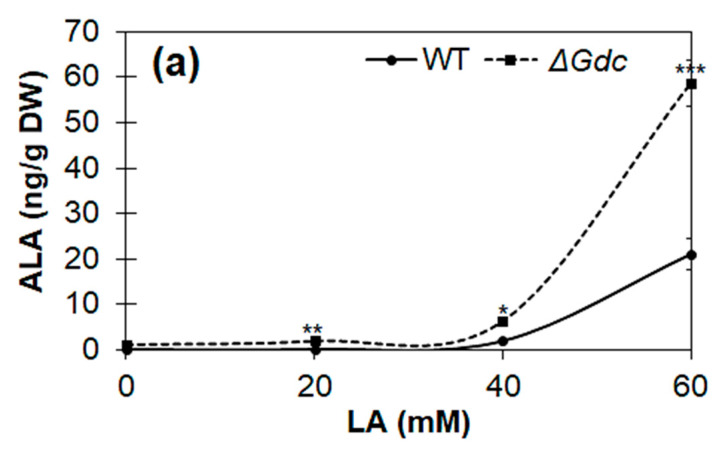
Quantification of the δ-aminolevulinic acid (ALA) production under a levulinic acid (LA) supply in the wild-type (WT) and the mutant Δ*Gdc* strains of *Synechocystis*. Effects of LA on (**a**) ALA production and (**b**) on the growth of cultures (biomass levels) in WT and Δ*Gdc Synechocystis* strains. Strains were grown until they reached the late-log phase, followed by a 48 h LA supplementation at the indicated concentrations. Results represent the mean ± standard deviation (SD) of experiments conducted in triplicate (*n* = 3). The differences between means of the individual groups were analysed using one-way analysis of variance (ANOVA) via GraphPad Prism 9.3.1 software (San Diego, CA, USA). * *p* < 0.05, ** *p* < 0.01 *** *p* < 0.005, **** *p* < 0.001 versus WT group. (**c**) Effect of the supplementation of various nutrients (10 mM of glucose and glutamate) in addition to 60 mM LA on ALA production in Δ*Gdc* strain cultures. The mutant strain was grown until the late-log phase and was subsequently supplied with LA + glucose or glutamate for 48 h. A culture growing without the supplemented nutrients but with the addition of LA, which was used as a control. Results represent the mean ± standard deviation (SD) of experiments conducted in triplicate (*n* = 3). The differences between means of the individual groups were analysed using one-way analysis of variance (ANOVA) via GraphPad Prism 9.3.1 software (San Diego, CA, USA). **** *p* < 0.001 versus control group.

**Table 1 ijms-24-01213-t001:** Oligonucleotides used in this study and their relative IDs as registered in the Cyanobase (http://genome.microbedb.jp/cyanobase/GCA_000009725.1, accessed on 7 April 2022); restriction sites are underlined.

Name	Target Sequence ID	Sequence 5′→3′	Purpose of Primer	Amplified Fragment Length (bp)
gdcF-KpnI	sll1641	GCTCTAGAATGGTGCATAAAAAAATTG	Forward primer for *Gdc*	1415
gdcR-KpnI		CTCGAGATGGCTAAAGTGGGA	Reverse primer for *Gdc*	
KanKp-F	*N.A.*	GGTACCAAGCCACGTTGTGT	Forward primer for the Kan^R^ cassette interrupting *Gdc*	1220
KanKp-R		GGTACCTGAGGTCTGCCTCG	Reverse primer for the Kan^R^ cassette interrupting *Gdc*	
F-gts	sll0179	TATAACCTGGCAGTGGTGGTG	Forward primer used in the qPCR reaction for *Gts*	190
R-gts		CCCCGTCTCGCTTAGATAAT	Reverse primer used in the qPCR reaction for *Gts*	
F-gtr	slr1808	AAGCGCTAACCCATCTGC	Forward primer used in the qPCR reaction for *Gtr*	152
R-gtr		GGGAATATTGCCAGTTTCAGA	Reverse primer used in the qPCR reaction for *Gtr*	
F-gsa	sll0017	CTATGGTGGTCGGGAAGAA	Forward primer used in the qPCR reaction for *Gsa*	193
R-gsa		AGCCGCAGCTAGTAAACCAT	Reverse primer used in the qPCR reaction for *Gsa*	
F-pbs	sll1994	TATCCCTTGTTTGCCGTTC	Forward primer used in the qPCR reaction for *Pbs*	174
R-pbs		CGTGGCATCGGTATCCTTAT	Reverse primer used in the qPCR reaction for *Pbs*	
F-gdc	sll1641	AACGTCCAGGTTTGTTGGGAAA	Forward primer used in the qPCR reaction for *Gdc*	166
R-gdc		TGCCGTCAAAGGTGCTACCT	Reverse primer used in the qPCR reaction for *Gdc*	
F-gdh	slr0710	TGAAGTTGCTAATGGGCC	Forward primer used in the qPCR reaction for *Gdh*	179
R-gdh		CCTTCAGGCGATCGTTAACTT	Reverse primer used in the qPCR reaction for *Gdh*	

The sequences of the forward and the reverse primers used in the real-time PCR (qPCR) reactions for *Kgd* (sll1981), *GabD* (slr0370), and *16S* (rRNA *16S*) were those described in a previous study [44].

**Table 2 ijms-24-01213-t002:** Production of ALA by photosynthetic microbes.

Microorganisms	Carbon/Nitrogen Source	ALA Production (µM)	References
*Synechocystis* Δ*Gdc*	Glucose	0.055	This study
*Synechocystis* Δ*Gdc*	Glutamate	0.044	This study
*Chlorella vulgaris*	CO_2_	1400	[23]
*Agmemnellum quadruplicatum*	Glutamate	0.225	[15]
*Chlorella regularis*	Glucose and yeast extract	3860	[45]
*Anacystis nidulans*	Glutamate	0.38	[46]
*Rhodobacter sphaeroides*	Succinate and glycine	0.75	[46]
*Rhodopseudomonas palustris* KG31	Glutamate and acetate	180	[40]

**Table 3 ijms-24-01213-t003:** Strains and plasmids used in this study.

Genotype		Source or Reference
**Strains**		
*E. coli* DH5α	F-φ80lacZΔM15, Δ(*lac*ZYA-*arg*F)U169, deoR, recA1, *end*A1, *hsd*R17(rk-, mk+), *pho*A *sup*E44, λ-, *thi*-1, *gyr*A96, *rel*A1	Stored in lab
*Synechocystis* PCC6803	Wild-type, naturally transformable	Pasteur culture collection
Δ*Gdc*	*Synechocystis* with *Gdc* disrupted by the Kan^R^ cassette	*Synechocystis* strain generated in this work
**Plasmids**		
pGEM^®^-T easy	Cloning vector, Amp^R^, *lac*Z, 3′-A overhangs, *mcr*	Promega
pGdc	pGEM^®^-T easy vector containing *Gdc*	This study
pGdc*_kan_*	pGEM^®^-T easy vector containing *Gdc* disrupted by the Kan^R^ cassette	This study
pUC4K	Source of the Kan^R^ cassette	Amersham

Amp^R^, Ampicillin resistance; Kan^R^, kanamycin resistance; mcr, multiple cloning region.

## Data Availability

Not applicable.

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
