# Peer review of "A Non-functional γ-Aminobutyric Acid Shunt Pathway in Cyanobacterium Synechocystis sp. PCC 6803 Enhances δ-Aminolevulinic Acid Accumulation under Modified Nutrient Conditions"

_ijms, 2023, doi:10.3390/ijms24021213_

Round 1

Reviewer 1 Report

Reviewer report for paper entitled "Lack of a Functional γ-Aminobutyric Acid Shunt Pathway in Cyanobacterium Synechocystis sp. PCC 6803 Enhances δ-Aminolevulinic Acid Accumulation Under Modified Nutrient Conditions". This work is focused on the redesign of genetic system of cyanobacteria for over production of δ-Aminolevulinic Acid through manipulation of the biosynthesis pathway. This is work of high interest for both scientific and industrial societies. However, some improvements are required before the publication of this work as follows:

- The introduction is well written and delivered all background related to this work. However, I recommend to highlight more the research gap in more practical approach providing the production achieved by other authors. 

- The results are well structured. However, some improvements are needed as follows:

- For figure 3, the Y axe need to be to be presented in w/w NOT w/number of cells (i.e. Chlorophyll a content (ug/mg cells) NOT cell number.

- Figures 4(a,b), many notes in the graph are not readable, need to provide it in larger size and bigger font. 

- Figure 6a need to be redraw. Consider the following. Growth need to be presented in mg or g cells per L as the other Y axe ALA(ng/g DW). The two upper and lower X axes are very confusing. I recommend to split this figure into two graphs linked to each others. 

- Authors need to provide kinetic table as comparison between the wild type strain and genetically modified one to compare (cell dry weight, growth rate, production (both volumetric and specific), and production rate) and need to compare these values with previously published work. 

- In materials and Methods part. Complete information about the source of WT strain need to be provided (full address of the culture collection)., i.e. (City, Country). For all equipment used should provide (Model, Company, City, Country). The same for all chemicals used (Cat. No., Company, City, Country). 

- Complete details of the cultivation system (cultivation vessel) used in this study. Need also to provide photo if available. 

- Conclusion part. Researchers need to be careful when write "is first-timely report in this study" should use the word "the best of our knowledge" when write such statement. Need also to highlight the potential application of the obtained results in biotechnology industries. 

Reviewer 2 Report

This manuscript will be of interest to IJMS readers. Especially transcriptional regulation of mutant strain for δ-Aminolevulinic Acid production along side abiotic factors. However, the study still needs a revision as there are plenty of typing errors to be improved. 

·        Title should be revised

·        There are some language mistakes throughout, which should be corrected (such as name of microbes are not Italic)

·        There are some formatting errors, should be carefully rechecked

·        Abstract is seeming to be very general. It is suggested to include quantitative results such as ALA production and chlorophyll-a with its unit.

·        As most of results presented in micrographs while values are presented in folds by comparing with control, it will be more appropriate for researchers if results should be numerical presented for reproducibility and comparison.

Other comments are furnished below:

1.      What was the reason for the selection of Synechocystis sp. PCC 6803? This strain was imported from France, don’t you think that the indigenous strains will be more suitable as it already adapted to environment?

2.      Page 1, Line 17: Write the cold temperature and salinity stress numerically with proper unit

3.      Page 1, Line 17: ‘several fold”. It is recommended to provide quantitative (numeric) results.

4.      Page 3, Line 107-113: It is recommended to delete this part as this should be part of material and methods section and not appropriate for result and discussion section.

5.      Page 3, Line 121-122: Properly write the amount of chlorophyll-a in three stages for audience to compare their results with your studies properly

6.      Page 5, Line 163-166: “There are several reports…extreme temperature”

In this statement authors claimed different environmental factors to improve ALA however, the cited reference only discussed about salinity. Authors are advised to recheck this and revise it.

7.      Page 7, Figure 4 (a): The amount of ALA in WT under different stress was to low in micrographs to not easily seen. It is recommended to change the y-axis value for proper illustration.

8.      Page 11, Section 3.2: Transformation protocol is missing. Explain the approach/protocol for transformation of recombinant plasmids into Synechocystis sp. PCC 6803 with proper citation.

9.      Page 11, Section 3.2: Properly write manufacturer name for oligonucleotides construction in this section

10.   Page 12, Page 404: Why selection of colonies required 5 generations sub-culturing?

11.   Page 12, Section 3.2; Page-13, Section 3.4: Kindly cite the reference for qPCR or proper details of manufacturer kits used.

12.   Page 13, Line 420: Culture volume of Synechocystis sp. PCC 6803 to produced δ-Aminolevulinic Acid was too small. It will be recommended to culture the strains at large scale. As the culture volume effects growth of cyanobacteria and microalgae so it will be suitable for elucidation of δ-Aminolevulinic Acid on large scale.

13.   Include a table to compare this study with closely related studies.

14.   Have the authors studied the impact of gene deletion on total carbohydrates, proteins and lipids content of cells and its comparison with wild cells?

15.   Was the culture supplied with an external CO2 source?

16.   The authors should expand the future prospects in conclusion section based on this study.

17.   Authors should also highlight the finding from the study for the wider commercial implications.

Round 2

Reviewer 1 Report

The revised manuscript showed significant improvement in the content. In my opinion, the only thing need to be corrected before publication as follows: 

Line 398 "in 500 ml cotton plugged conical flasks with a total volume of 200ml" Need to be changed to:

"in 500 cotton plugged Erlenmeyer flasks with a working volume of 200ml"

This because the total volume of the flask as written is 500ml and 200ml should be the working volume NOT (total volume). 

This is very minor correction which need to be consider before printing the manuscript. But in the current status, the manuscript of high quality and I recommend to accept it in the current form after doing this minor correction. 
